# Effects of Halogen Lamp and Traditional Sun Drying on the Volatile Compounds, Color Parameters, and Gel Texture of Gongliao *Gelidium* Seaweed

**DOI:** 10.3390/foods12244508

**Published:** 2023-12-17

**Authors:** Wen-Chieh Sung, Hong-Ting (Victor) Lin, Wei-Chih Liao, Mingchih Fang

**Affiliations:** 1Department of Food Science, National Taiwan Ocean University, Keelung 202301, Taiwan; sungwill@mail.ntou.edu.tw (W.-C.S.); hl358@ntou.edu.tw (H.-T.L.);; 2Center of Excellence for the Oceans, National Taiwan Ocean University, Keelung 202301, Taiwan

**Keywords:** sun drying, halogen lamp drying, *Gelidium* seaweed, jelly, volatile compounds, dried seaweed

## Abstract

Traditionally, the processing of *Gelidium* seaweed into *Gelidium* jelly was very complicated, and involved repeated washing with water and sun drying for seven rounds. The seaweed, which is originally reddish-purple in color, turns yellow in color after the repeated washing and sun drying cycles. However, the sun drying process can only be used on sunny days. Therefore, this study evaluated an alternative method, halogen lamp drying, and compared the qualities of the product, *Gelidium* jelly, made using the halogen lamp drying and traditional sun drying methods. The properties investigated included the agar yield, gelling temperature, hardness, springiness, rheological parameters, sensory attributes, color, and volatile compounds. The results demonstrated that the halogen lamp drying method required 12 washing and drying cycles to achieve similar jelly properties to seven rounds of sun drying in the experimental conditions. Volatiles including heptanal, β-ionone, and (E)-2-decenal could be used as indicators to monitor the washing and drying processes. Halogen lamp drying could be an alternative processing method for seaweed drying, especially on rainy days.

## 1. Introduction

Seaweeds generate various low-molecular-weight organic volatiles which are essential for odor and sensory perception, and they are highly related to the physiology of seaweed species. The volatile compounds contained in seaweeds, including acids, alcohols, aldehydes, esters, hydrocarbons, ketones, phenols, terpenes, and halogenated and sulfur compounds, are sometimes unacceptable for some consumers [1]. Therefore, most seaweeds need to be processed (mostly by drying) prior to consumption, and still remain only a minor portion of the diet in Western countries.

Dried red seaweed *Gelidium amansii*, J. V. *Lamouroux*, is the raw material of *Gelidium* jelly, which is a gelled product acquired by boiling dried red seaweed in water for 1 h. The extracted agar solution is then filtered and cooled to room temperature to form *Gelidium* jelly [2]. The jelly may be added into ice cold honey water as a popular summertime beverage. The drying process of *Gelidium amansii* is believed to require repeated washing and sun drying, for seven rounds, in order to obtain the desired gel strength for the traditional soft agar pudding. New Taipei City is the main area producing dried red seaweed in Taiwan. However, sunshine hours (when sunshine is greater than 200 W/m^2^) are limited, amounting to 1317–1430 h/year, and annual rainfall is high, being between 2744 and 3590 mm/year, in the Gongliao district (25°01′ N and 121°54′ E) [3], which limits the production of dried red seaweed. Traditionally, the drying process starts with evenly piling red seaweed on the ground, forming a seaweed mat approximately 1 cm in height [4]. The impurities and other seaweeds are removed, and the seaweed is dried under the sun. The seaweed mat is then rolled up in the evening and washed thoroughly with water. The next day, the seaweed is rolled out under the sun to dry again, and washed with water again in the evening. The washing and drying process is repeated seven times in seven days, if the weather allows. The traditional solar drying method is a very convenient and cheap processing method in summertime, compared to other food drying methods—if the sunshine is strong enough. This process could remove seaweed’s fishy odor and turn the reddish-purple color into yellow [4]. The reddish-purple matter of red seaweed is susceptible to photochemical degradation under solar radiation [5].

Information on the volatile compounds of dried *Gelidium* seaweed, which contributes to its sensory profile, is rare. This study investigated the volatile changes during the washing and sun drying cycles of *Gelidium* seaweed. Sun light is important to the drying process of *Gelidium* seaweed because it provides a bleaching effect. An alternative drying method by halogen lamp instead of sun drying was studied for first time in the literature. The jelly properties and volatiles of the two drying methods in terms of *Gelidium* seaweed were compared, described, and discussed.

## 2. Materials and Methods

### 2.1. Materials

Red seaweed *Gelidium amansii* (J. V. Lamouroux) was collected from March 2022 to July 2022 on three occasions under low tide conditions in the Gonglian district (25°01′ N and 121°54′ E) of New Taipei City, Taiwan. The fresh red seaweed was kept in a freezer at −18 °C until use. C7–C30 saturated alkanes and n-hexane were purchased from Sigma Aldrich (St. Louis, MO, USA).

### 2.2. Dried Red Seaweed Preparation

Traditionally, the red seaweed is washed and dried under sun light, then collected in the evening. The next day, the process is repeated for 7 cycles over 7 days. In the current study, the traditional sun light drying method was simulated. Five hundred grams frozen seaweed for each batch was placed under running tap water until completely thawed to tap water temperature (25 °C). Then, the seaweed was evenly placed on a stainless mesh (18 × 10 cm, L × W) to a thickness of about 1 cm. The red seaweed was dried under sun for 2 h (the luminance was around 35,000 to 50,000 lux) or dried under a 300 W LED halogen lamp (18 × 16 × 11.5 cm, L × W × H) (Anjia Light Store, Kaohsiung) for 2 h (the average luminance was 15,000 to 20,000 lux) at a temperature of around 60 °C, placed at the center of the algae during the drying process. After the drying process, the red seaweed was stored at night. The next day, the red seaweed was washed and dried under the sun for 3 and 7 cycles (sun drying), dried by halogen lamp for 7, 9, and 12 cycles (halogen lamp drying), respectively. The additional drying cycles under halogen lamp drying (up to 12 cycles) made the gel harder. An oven-dried red seaweed (50 °C for 4 days) reaching a moisture content of 10–13% was used as a control sample. Another red seaweed sample was dried in an oven with exposure to a UV-C light at a distance of 30 cm from the sample surface (Model A-01, 253.7 nm; 30 W, PJLink, Taipei, Taiwan) to simulate sun light for the oven drying process.

### 2.3. Moisture and Agar Content

The moisture content of dried red seaweeds was measured according to AOAC method 984.25 by an oven at 105 °C [6]. Agar content was measured by boiling 2 g of seaweed each in deionized water (200 g) for 60 min. Then, the extracted agar solution was filtered, and the filtrate was lyophilized under vacuum for 48 h using a freeze dryer (LABCONCO 12 L, Kansas City, MO, USA). The yield of agar was expressed as % *w*/*w* on the dry basis of 2 g dried seaweed sample. All tests were performed in triplicate.

### 2.4. Seaweed Jelly Texture Analysis

The above extracted and filtered agar solution each was immediately poured into a glass beaker (diameter of 38.0 mm and height of 38.0 mm). The beaker was stored at 4 °C overnight to form jelly. Hardness (N) and springiness (%) of jelly sample was measured using a texture analyzer (TA-XT2, Stable Micro Systems, Godalming, UK) equipped with a 4 mm diameter cylinder probe (P/0.5) (test speed: 1.0 mm/s; target distance: 15.0 mm; trigger force: 15 g) according to the methods of Hurler et al. [2]. Springiness was defined as the percentage (%) of the second compression force to the maximum compression force (hardness) during two compression cycle. Springiness represented the hardness of the sample at second bite. All tests were performed with 3 replicates.

### 2.5. Color Analysis

Dried red seaweed sample each was ground into powder with a pulverizer (D3V-10, Yu-Chi Machinery Co., Ltd., Chan Hua, Taiwan). The CIELAB color expressed by L* (lightness), a* (redness/greenness), b* (yellowness/blueness) color scale of seaweed powder was conducted with a spectrocolorimeter (TC-1800 MK-II, Tokyo, Japan). CIE D65 illuminant and 10° Standard Observer were used. Standardization was performed with a black cup and a white tile (X = 79.2, Y = 80.7, Z = 90.7). The seaweed powder was loaded onto a quartz sample cup for each sample and triplicate were recorded per treatment. The color difference ΔE was calculated with the following formulation: ΔE = [(ΔL*)^2^ + (Δa*)^2^ + (Δb*)^2^]^1/2^(1)
ΔL* = L*_sample_ − L*_fresh_(2)
Δa* = a*_sample_ − a*_fresh_(3)
Δb* = b* _sample_ − b*_fresh_(4)

### 2.6. Rheological Measurement

The dynamic viscoelastic properties of filtered agar solution were measured using a rheometer (Physica MCR 92, Anton Paar Gmblt, Ostildern, Germany) at a temperature range of 90 °C to 20 °C, with a decreasing temperature of 2 °C/s. The filtrate of the boiling dried seaweed 0.56 mL was immediately poured on the rheometer’s lower plate (PP252, which was conditioned at 90 °C). The parallel plate was lowered to the solution, the measurement was initiated after thermal equilibrium. Each run consisted in three steps. The first gelation step conducted at an amplitude strain of 0.5%: temperature ramp from 90 °C to 20 °C at a cooling rate of 2 °C/s and an angular frequency of 30 rad/s. The second curing step: time sweep of 1800 s, at 20 °C and an angular frequency of 30 rad/s at shear strain of 0.5%. The third mechanical spectra step: frequency sweep from 1 to 100 rad/s (ramp linear) at 20 °C and 0.5% shear strain. All the rheological measurements were analyzed using the Rheoplus software (version 3.21, Anton-Paar) to generate elastic modulus (G′), viscous modulus (G″), and derived parameters. A strain amplitude sweep (from 0.01% to 100%, at 1 rad/s and 20 °C) was applied immediately after each run to verify the measurement within the linear viscoelastic range [7].

### 2.7. Volatiles of Dried Seaweed

The dried red seaweeds from different washing and drying cycles were cut into 0.5 cm pieces. A sample of 0.5 g was placed into a 20 mL headspace vail, and 12 mL water was added. Ethyl caprate 4 μL at concentration of 20 ppm in methanol was added as internal standard. The headspace vail was sealed with a Teflon-lined septum and screw cap, which was conditioned at 70 °C for 30 min. A 30/50 μm DVB/CAR/PDMS fiber was then inserted into the headspace vial for 30 min as absorption process [8]. Then, the fiber was inserted into the GC injector for 5 min using the splitless mode (vent time for 3 min). The injector was set at 250 °C. Volatile compounds were separated and analyzed using a gas chromatograph (GC, 7890B, Agilent Tech., Santa Clara, CA, USA) equipped with a mass spectrometer (MS, 7000C, Agilent Tech.) and a DB-5MS column (30 m × 0.23 mm, film 0.15 μm, Agilent Tech.). Helium was the carrier gas at a flow rate of 1.0 mL/min. The oven temperature was kept at 40 °C for 3 min and then increased to 70 °C at a rate of 7 °C/min. Then, the oven temperature was maintaining at 70 °C for 3 min followed by a further increase to 170 °C at a rate of 10 °C/min, and an increase to 230 °C at a rate of 20 °C/min. The final holding time was 10 min. The MS scanning range was *m*/*z* 30–450.

Volatile compounds were identified based on their retention indices (RI), which were calculated using n-alkanes (C7-C30, Sigma Aldrich, St. Louis, MO, USA) [9], and matched MS spectra library (NIST 17). Semiquantitative analysis was performed by the relative area of the compounds to internal standard, ethyl caprate. The estimated odor activity values (OAVs) of selected volatiles were calculated by dividing the concentrations (semiquantitative data) with the odor thresholds (OTs) from references. 

### 2.8. Sensory Evaluation

Sensory evaluation of *Gelidium* jellies made from dried seaweed by various drying methods were carried out by untrained panelists on the basis of preference tests. The panelists consisted with 37 females and 29 males in age between 19 and 85 were recruited. Attributes on appearance, color, flavor, texture, overall acceptability (liking), and fishy odor were selected as the important qualities of *Gelidium* jellies and asked [7]. Samples were coded with 3 random digits and supplied to the panelists in random order. Panelists were instructed to evaluate each of the above-mentioned attributes by ranking from “1 = extremely dislike” to “9 = extremely like”. A nine-point hedonic test for fishy odor (1 = no fishy smell, and 9 = strong fishy smell) was also conducted. 

### 2.9. Statistical Analysis 

All tests were performed in triplicate. One-way analysis of variance (ANOVA) and Turkey’s tests at a 5% significance level (*p* < 0.05) were analyzed with the IBM SPSS statistics (Version 23.0, IBM, Armonk, NY, USA). The General Linear Model (GLM) was assessed for one-way analysis of variance in SPSS. Pearson correlation analysis was used for the examination of linear correlations on gel appearance, texture, hardness, color parameters, fishy odor, and overall acceptability at significance levels of *p* < 0.05 and *p* < 0.01. The semiquantitated contents of volatiles in dried seaweeds by different washing and drying cycles were used as variables for principal component analysis (PCA) and hierarchical cluster analysis (HCA). Visualized graphs were generated by MetaboAnalyst 5.0 (http://www.metaboanalyst.ca/, accessed on 20 August 2023). 

## 3. Results and Discussion

### 3.1. Effects of Halogen Lamp Drying, Sun Drying, and Washing Cycles on the Agar Yield and Properties

The agar yields of red seaweed obtained by different drying processes and cycles are shown in Table 1. The agar yields ranged from 23.0% to 46.0%. Sun drying for 7 cycles showed the best agar yield, whereas oven-dried seaweed obtained the lowest agar yield probably because repeated washing and sun drying would damage the cell wall and enhance the agar extracting yield. Nevertheless, the energy required for halogen lamp drying was less than that required for sun drying. Therefore, the maximum yield of agar was observed during washing and halogen lamp drying for 9 cycles (Table 1). However, further washing and halogen lamp drying might leach some available soluble polysaccharides into washing water, which decreased the agar yield after 12 treatments. The main components of *Gelidium* seaweed are hydrophilic polysaccharides with water surrounded in the internal and external substructure by hydrogen bonding. The washing process mainly hydrated the dried seaweed, eased the availability of soluble polysaccharides, and removed impurities. The agar yields from native and alkali-modified extractions of the red seaweed *Gracilaria* lemaneiformis were reported to be approximately 29.7% and 25.4%, respectively [10]. Such yields are influenced by various parameters such as soaking time, soaking temperature, water-to-seaweed ratio, boiling temperature, and duration [11]. Other pretreatments, such as acid hydrolysis and ultrasound-assisted extraction, were found to increase not only the agar yield but also some valuable biomolecules at a shorter processing time [12].

The qualities of agar were altered by seaweed growth under different conditions such as light availability, salinity, and seawater temperature [13,14]. *Gelidium* sp. was reported to provide better agar quality with regard to gel strength compared with other species such as *Gracilaria* [15]. *Gracilaria* species exhibited low agar quality because of their high sulfate (L-galactose-6-sulfate) content, which could be treated with alkali and converted into 3,6-anhydro-L-galactose to increase gel strength [11]. The agar jellies from different drying methods and cycles showed low gelling temperatures in the range of 26 °C–31 °C, which was lower than that from *G. lemaneiformis* in the range of 40 °C–41 °C [10]. The jelly made from oven-dried *Gelidium* seaweed (F, control) showed the lowest gelling temperature (26 °C), whereas the gel extracted from sun-dried seaweed with seven times exposure under sun light displayed the highest gelling temperature (Table 1). According to the US Pharmacopoeia, the gelling temperature of commercial agar should fall between 34 °C and 43 °C. Agars from *Gracilaria* species were reported in the gelling temperature range of 40 °C–52 °C [10]. In this study, the gelling temperature of *Gelidium* was found to be low, ranging from 26 °C to 31 °C. The high gelling temperature reported by Li et al. [10] might be due to the agar concentration, which applied 2 g of the dried *Gracilaria* seaweed into less deionized water (100 mL), and longer extraction time (2 h). In this study, agar was extracted by boiling 2 g of *Gelidium* seaweed in 200 mL of deionized water for 1 h.

Gel hardness, an important gel texture characteristic of consumer acceptance, is shown in Table 1. The hardness of the jelly samples ranged from 0.23 N (F) to 3.54 N (S7). The jelly hardness of oven-dried *Gelidium* seaweed (F) was lower than that of sun-dried and halogen lamp-dried red seaweeds. This result might be due to the fact that the water-soluble agar was still held inside the cell wall during oven drying without washing and sun drying cycles. This study found that the color of red seaweed did not change (i.e., no photobleaching occurred) when subjected to 45 °C–105 °C oven drying. Later, UV-C light irradiation was applied during oven drying. However, the color of red seaweed did not change. A full band or a certain frequency band might be required for photobleaching, but such a band was not in the scope of this study. The jelly of sun-dried seaweeds with seven times sun light exposure obtained the highest jelly hardness (Table 1), followed by the jelly of sun-dried seaweeds with three times exposure and halogen lamp-dried seaweeds with 12 times exposure. Therefore, the effect of sun drying and washing will increase the agar yield and jelly strength of agar during extraction.

All seaweeds were dried under the sun or a halogen lamp for 2 h, reaching a moisture content of approximately 10–13%. The effect of halogen lamp drying (15,000–20,000 lux) and washing for 12 times showed lower agar yield compared with sun drying for seven times because the photobleaching effect on the red seaweeds by halogen lamp drying was not effective as sun drying (35,000–50,000 lux). The appearance of seaweed and agar is shown in Figure 1. The maximum agar yield was observed at L9 (washing and halogen lamp drying cycles for nine times) and decreased at L12. The jelly strength increased gradually with sun and halogen lamp drying cycles (Table 1). The solar spectral irradiance for photobleaching in the first 5 h has proven to increase agar gel strength but not agar yield [16]. The gel strengths of native *G. lemaneiformis* and *G. asiatica* processed by oven drying at 60 °C were reported to be 2.66 and 4.56 N, respectively [10]. Li et al. [10] extracted agar by boiling 10 g of *Gracilaria* seaweed in deionized water (500 mL) for 2 h. However, in this study, agar was extracted by boiling 2 g of *Gelidium* seaweed in 200 mL of deionized water for 1 h. Therefore, the gel strength of *G. asiatica* was 4.56 N, which was higher than that (3.5 N) of *Gelidium* seaweed sun dried seven times. This result may be due to the fact that more dried seaweed sample weight and longer extraction time were used to extract agar. The gel strength depends on the chemical structure of the extracted agar and agar concentration, and it is inversely proportional to the sulfate content of the agar [10].

The springiness of *Gelidium* jellies showed a similar trend to the hardness. The jelly from seaweeds sun dried for seven cycles showed the highest springiness, followed by the jelly from seaweeds sun dried for three cycles and from seaweeds halogen lamp dried for 12 cycles (Table 1). Table 1 shows that *Gelidium* jelly made from the oven-dried sample (control, F) had the weakest gel hardness, which exhibited a liquid-like characteristic.

### 3.2. Effects of Drying Methods and Washing Cycles on Rheological Parameters of Agar Solutions Extracted from Gelidium Seaweeds

The strain sweep tests demonstrated that the linear viscoelastic range of all extracted agar jellies was between 0.01% and 100% at 1 rad/s and 20 °C, confirming that all measurements conducted at 0.5% shear strain was within the linear viscoelastic range. Values of storage modulus (G′) were used to compare the different treatments and were considered as an indicator of gel rigidity [17]. The gelation of the agar sol–gel transition of dried red seaweed extract occurred during the formation of junction zones, including cross-links and aggregation of helices [18]. In addition, the gelation of pure agarose occurred in the first hardening step at 45 °C during cooling from the solution state at 90 °C. In this study, the hardening step of cross-link gelation was not observed at 45 °C (Figure 2A). Loss moduli (G″) were greater than storage moduli (G′), indicating that the hot agar extracted solutions were in a liquid state at 45 °C. Nevertheless, the second step of the helices joining to form aggregates was observed below the sol–gel transition temperature ranges (31 °C–26 °C). Further hardening at 20 °C likely corresponded to the aggregation of helices, which further increased the modulus G′, especially for *Gelidium* jelly S7. The storage modulus of S7 showed the highest value, followed by L12 and S3. These storage modulus results were consistent with the hardness of *Gelidium* jellies (Table 1 and Figure 2A). The storage modulus (G′) was greater than the loss modulus (G″) at 20 °C, which indicated a typically viscoelastic solid behavior because of the contribution of *Gelidium* jellies.

The storage modulus of curing curves (a relaxation of stresses developed during gel formation of *Gelidium* jellies at 20 °C for 30 min) is shown in Figure 2B. The storage modulus (G′) of L12 decreased after 5 min of aging at 20 °C, which indicated that the agar jelly network was unstable and that the behavior under stress was not in equilibrium even after aging for short periods compared with the other samples. All jellies could be tenuous gel-like networks, which were weak gels. In particular, the L12 jelly was easily broken when subjected to high stress (Figure 2B). Figure 2C shows the mechanical spectra of *Gelidium* jellies. The storage modulus of fresh and untreated (F) and L9 jellies showed a typical behavior of weak gels compared with the storage modulus of S7, S3, L7, and L12 samples, which showed dramatic increases in the frequency sweep for the dynamic moduli. The values of slope were indicative of the distinct nature of bonding in the network structures of jellies. Steeper slopes indicated a liquid-like characteristic of the sample at a high frequency sweep [17,19], such as F and L9 *Gelidium* jellies. This finding indicated that the jelly structures of the F and L9 groups were deformed during dramatic frequency sweep changes. These two *Gelidium* jellies may retain their jelly structure under mild frequency sweep changes. Therefore, the F and L9 *Gelidium* jellies showed a liquid-like jelly.

### 3.3. Color Measurements of Dried Red Seaweeds and Appearance of Gelidium Jellies with Various Drying Methods and Washing Cycles

The color parameters of *Gelidium* seaweed with different drying methods and washing cycles are shown in Table 2. The L* value of S7 *Gelidium* seaweed had the highest L* and b* values and the lowest a* value. This result indicated that water washing and sun light exposure seven times achieved the highest levels of photobleaching in approximately 14 h during midday of around 35,000–50,000 lux. The appearance of dried *Gelidium* seaweeds and *Gelidium* jellies is shown in Figure 1A and B, respectively. The color of dried seaweeds turned yellow because of the photobleaching effect, and this process was related to irradiance levels and high temperature [5]. Fresh *Gelidium* seaweed (F) dried in a 50 °C oven barely received any photobleaching. The control sample F showed the lowest L* and b* values in color (Table 2). In addition, jelly made from oven-dried seaweed still contained pigments and was dark red in color. Pigmental proteins, including allophycocyanin, C-phycocyanin, and R-phycoerythrin, are reported to be the main pigments of red seaweed and can be photobleached under irradiation of a mercury lamp (450 W) with visible light at wavelengths longer than 470 nm, which drives the generation of superoxide radical anion, hydrogen peroxide, hydroxy radical, and singlet oxygen in oxygen-saturated aqueous solutions [20]. In this study, all treatments received either sun drying or halogen lamp drying, and they showed a bleached effect and turned yellow with extended irradiation. Quintano et al. demonstrated that the red alga *Gelidium* corneum would bleach under different sun irradiation levels [5].

### 3.4. Volatile Compounds of Dried Gelidium Seaweeds

Volatiles analyzed by GC-SPME of dried *Gelidium* seaweeds subjected to different washing and drying cycles are provided in Appendix A. These volatiles comprised aldehydes, ketones, and alcohols. The total volatiles semiquantitated in dried red seaweeds were listed in descending order: S3 (3387 ng/g), L7 (3232 ng/g), L9 (2807 ng/g), S7 (2729 ng/g), L12 (2441 ng/g), and F (2119 ng/g). Prolonged washing and drying in either sun drying or halogen lamp drying favored the evaporation of volatiles. Among all compounds identified, hexanal, 2-hexenal, heptanal, octanal, 2-nonenal, and β-cyclocitral were found in all dried seaweed samples. These compounds were responsible for the green, floral, fatty, and seaweed odor characteristics.

The odor activity value (OAV) was calculated from the semiquantitative concentrations of volatiles divided by the odor threshold (OT). Higher OAV implied a high contribution to the flavor profile of dried seaweed aroma (Table 3). Aldehydes in seaweeds primarily result from the enzymatic action of lipoxygenases, auto-oxidation, or degradation of polyunsaturated fatty acids [21]. Despite their higher thresholds compared with other volatile compounds, aldehydes contribute remarkably to the aroma profiles of many foods. In this study, 17 aldehydes were identified in dried red seaweeds, accounting for approximately 80% of the total volatile compounds. Among them, 2,4-nonadienal and 2,4-decadienal provided extremely high OAVs in all sun-dried and halogen lamp-dried seaweed samples (Table 3). Therefore, these two aldehydes contribute to the aroma of dried *Gelidium* seaweeds by providing fat, wax, green, and fried (oily) odors.

The OAV of 2-decenal significantly increased after drying and washing. Saturated aldehydes typically release green, grassycitrus lemony, or orange peel-like odor sensations, as well as pungent, fatty, soapy, or tallowy aroma notes. Hexanal, with its grassy and green apple scent, is produced by lipid oxidation or by the sequential action of lipoxygenase/fatty acid hydroperoxide lyase on linoleic acid [22]. Hexanal contributes to the grassy and fishy odor in seafood. The concentration of hexanal was higher in oven-dried *Gelidium* seaweeds, which did not undergo washing. 2-Nonenal and 2,4-decadienal represented a stale beer aroma with odor thresholds of 0.1 and 0.07 ng/mL, respectively. Beer brewing industries are required to monitor the concentrations of these two compounds because they contribute to the off-flavor in beer. These two compounds were also the primary volatile components of bread crumb, providing roasted and green flavor notes. They were increased in dried seaweeds with prolonged washing and drying cycles. Beta-cyclocitral, α-ionone, and β-ionone were generated from the oxidative cleavage of α- and β-carotene at the double bond site between carbons 7 and 8 [23]. Although they were not considered as important odor contributors in dried seaweeds because of their lower abundance, they were reported as important odorants in raw seaweeds. Alpha and β-ionone were high in fresh, S3, and L7 samples but were not detected in samples that underwent prolonged drying, such as S7, L9, and L12, because of photodegradation by UV photolysis during sun drying and halogen lamp drying [24,25]. Kim et al. [24] demonstrated that β-ionone could be partially removed by UV photolysis.

**Table 3 foods-12-04508-t003:** Odor active values of dried *Gelidium* seaweeds analyzed by GC-SPME-MS.

				OAV **							
	Compound	OT * (ng/mL)	F	S3	S7	L7	L9	L12	Odor	Odor Reference	OT Reference
1	hexanal	4.1	69	43	34	57	38	35	green, grass	a	[26]
2	2-hexenal	17	4	-	2	2	2	1	green, leaf	a	[26]
3	2-heptanone	1	-	-	-	12	5	-	Soap	a	[26]
4	heptanal	3	167	31	7	12	7	5	fat, citrus, rancid	a	[26]
5	2-heptenal	4.2	-	80	42	67	59	-	fat, grass	a	[26]
6	1-octen-3-one	0.05	-	-	-	885	348	147	mushroom, butter	a	[26]
7	1-octen-3-ol	14	7	-	-	2	1	-	mushroom	a	[26]
8	octanal	1.4	168	210	205	121	99	100	fat, soap, green	a	[26]
9	2-octenal	3	-	102	21	79	60	37	green, nut, fat	a	[26]
10	1-octanol	42	-	5	1	3	1	-	pulpy, fruity, sweet	[27]	[26]
11	3,5-octadien-2-one	0.15	-	-	-	299	60	-	geranium, metal	a	[26]
12	3,5-octadien-2-one	150	-	1	-	-	-	-	earth, must	a	[26]
13	nonanal	1	-	332	253	293	287	305	fat, citrus, green	a	[26]
14	2,6-nonadienal	0.09	481	316	-	159	53	-	cucumber, wax, green	a	[26]
15	2-nonenal	0.1	836	983	1370	778	1612	2147	cucumber, fat, green	a	[26]
16	decanal	0.1	-	362	464	209	379	595	soap, tallow	a	[26]
17	2,4-nonadienal	0.09	-	2057	1206	4158	3364	2939	fat, wax, green	a	[26]
18	2-decenal	1	29	206	646	112	475	527	tallow	a	[26]
19	β-ionone	0.03	12,632	747	-	482	-	-	violet, flower	a	[26]
20	2,4-decadienal	0.07	-	2558	2154	4067	2634	1806	fried, wax, fat	a	[28]
21	undecanal	0.4	-	-	140	17	53	119	oil, pungent, sweet	a	[26]
22	2-undecenal	3.5	-	18	30	-	6	27	Sweet	a	[29]
23	dodecanal	0.5	-	50	124	47	73	87	fatty, green	[30]	[26]
24	α-ionone	0.6	424	106	-	60	20	20	wood, violet	a	[26]
25	tetradecanal	14	-	2	1	-	-	-	Aldehyde	[31]	[28]

* OT: odor threshold. ** OAV: odor activity value. a: Flavornet and human odor space. F (control): Oven-dried *Gelidium* seaweed; S3: Washing and sun drying cycles for 3 times; S7: Washing and sun drying cycles for 7 times; L7: Washing and halogen lamp drying cycles for 7 times; L9: Washing and halogen lamp drying cycles for 9 times; L12: Washing and halogen lamp drying cycles for 12 times. http://www.flavornet.org/flavornet.html (accessed on 30 May 2023).

### 3.5. Sensory Evaluation

The sensory attributes of *Gelidium* jelly, such as appearance, color, texture, flavor, overall acceptability, and fish odor, are summarized in Table 4. All processed seaweeds obtained higher scores, including the appearance of S3, S7, L9, and L12 and the color of S3, S7, L7, L9, and L12, than sample F (*p* < 0.05), which was dark purple in color and a weak liquid-like gel. Jellies from sun- and halogen lamp-dried seaweeds were light yellow in color (Figure 1B) and firm in texture, especially S7 and L12 jellies. The textures of samples S3, S7, and L12 were all acceptable with scores higher than 5 (average hedonic score). However, only the flavor of sample S7 was acceptable with a score higher than 5. The overall acceptability of samples S3, S7, and L12 were all acceptable with scores higher than 5 (average hedonic score). The fishy odor of samples F, L7, and L9 was detected with scores higher than 5 (average hedonic score showed dislike). In this study, the majority of panelists preferred *Gelidium* jellies with a harder texture, a color of light yellow, and lower fishy odor. Correlation analysis was conducted between the physicochemical characteristics and sensory attributes to comprehensively understand the relationship among the jelly quality properties of seaweeds with different treatments (Appendix A). The overall acceptability of *Gelidium* jelly samples was negatively correlated with fishy odor (*p* < 0.01) and positively correlated with the b* value of dried seaweeds, as well as the hardness and springiness of jellies (*p* < 0.05).

### 3.6. Principal Component Analysis of Volatiles 

The amounts of semiquantitated volatiles in each processed seaweed (Appendix A) were used as variables and visualized by principal component analysis (PCA, Figure 3A). PC1 and PC2 accounted for 85% of the variation. The triplicate samples of each drying treatment of *Gelidium* seaweeds were clustered together and separated into four different groups: the F sample group, L7 and S3 sample groups, L9 sample group, and L12 and S7 sample groups. The closer groups were related or called similar. Hierarchical cluster analysis was used to reveal the changes in the volatile compound profiles among all six samples after different drying and washing cycles. The resulting heatmap (Figure 3B) demonstrated similar results to PCA, that is, the effects of sun drying for 7 cycles on volatiles were similar to those of halogen lamp drying for 12 cycles, and the effects of sun drying for 3 cycles were similar to those of halogen lamp drying for 7 cycles. This study found that halogen lamp drying may be a good alternative for sun drying in the production of dried *Gelidium* for jelly. Halogen lamp drying can be applied, rain or shine, and this process was cleaner and hygienic. The quality of gel produced by halogen lamp-dried seaweeds was similar to that of traditional sun-dried seaweeds with regard to flavor, texture, and color. The power (watts) of the halogen lamp may be increased to speed up the required drying time.

## 4. Conclusions

The washing and drying cycles of *Gelidium* seaweeds by sun drying and halogen lamp drying were considered as significant parameters of the sensory quality and volatile composition of *Gelidium* jellies. This study reported for the first time that halogen lamp drying provided similar effects to sun drying, which improved the yield and strength of extracted agar. Washing and sun drying removed unpleasant smells (volatile components), such as hexanal, but enhanced long-chain unsaturated alkylaldehydes, such as decenal and 2-nonenal, which provided green, waxy, and floral odors. The novel introduction of halogen lamp drying for *Gelidium* seaweed could be an alternative for the traditional sun drying process, especially on rainy days. However, the drying and illuminant energy of the halogen lamp was not high enough in this study. It can be improved by a high-power halogen lamp or infrared lamp that would reduce the processing cycle times. Sensory evaluation and volatile odorant analysis demonstrated that *Gelidium* jellies prepared by traditional sun drying for 7 cycles and halogen lamp drying for 12 cycles showed similar results. This investigation and report may benefit the seaweed jelly industry by aiding the development of drying techniques and seaweed processing.

## Figures and Tables

**Figure 1 foods-12-04508-f001:**
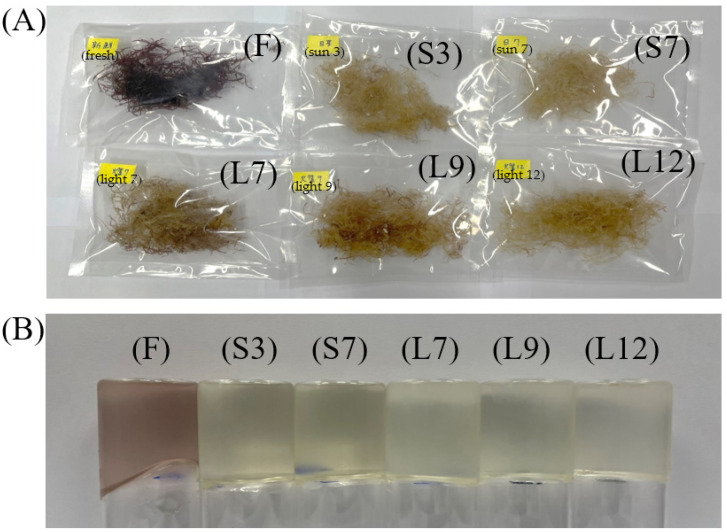
(**A**) Appearance of *Gelidium* seaweed; (**B**) Appearance of *Gelidium* jelly. F (control): Oven-dried *Gelidium* seaweed; S3: Washing and sun drying cycles for 3 times; S7: Washing and sun drying cycles for 7 times; L7: Washing and halogen lamp drying cycles for 7 times; L9: Washing and halogen lamp drying cycles for 9 times; L12: Washing and halogen lamp drying cycles for 12 times.

**Figure 2 foods-12-04508-f002:**
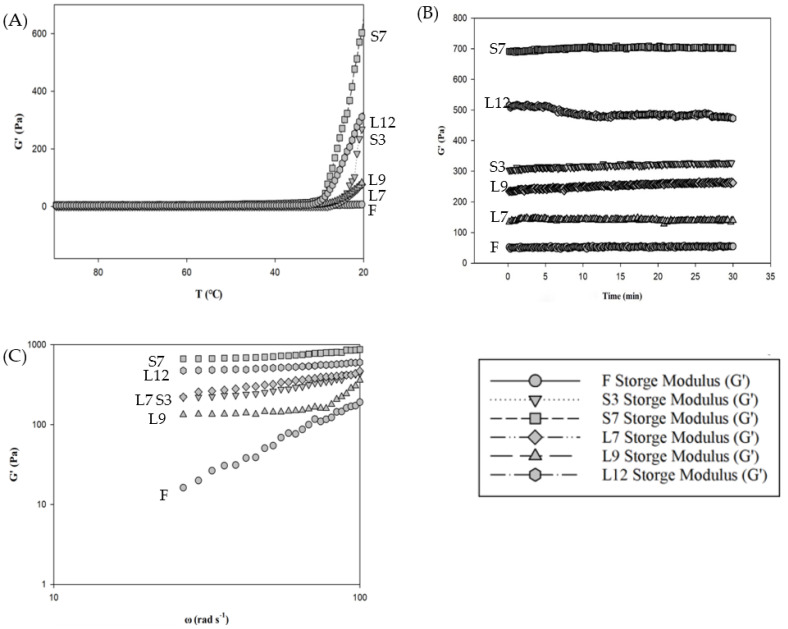
Analysis of rheological properties of *Gelidium* jellies: (**A**) Gelation: Storage modulus as a function of temperature, during cooling of *Gelidium* jellies. (**B**) Curing: Storage modulus as a function of time during ageing of *Gelidium* jellies. (**C**) Mechanical spectra: Storage modulus of *Gelidium* jellies as a function of frequency. F (control): Oven-dried *Gelidium* seaweed; S3: Washing and sun drying cycles for 3 times; S7: Washing and sun drying cycles for 7 times; L7: Washing and halogen lamp drying cycles for 7 times; L9: Washing and halogen lamp drying cycles for 9 times; L12: Washing and halogen lamp drying cycles for 12 times.

**Figure 3 foods-12-04508-f003:**
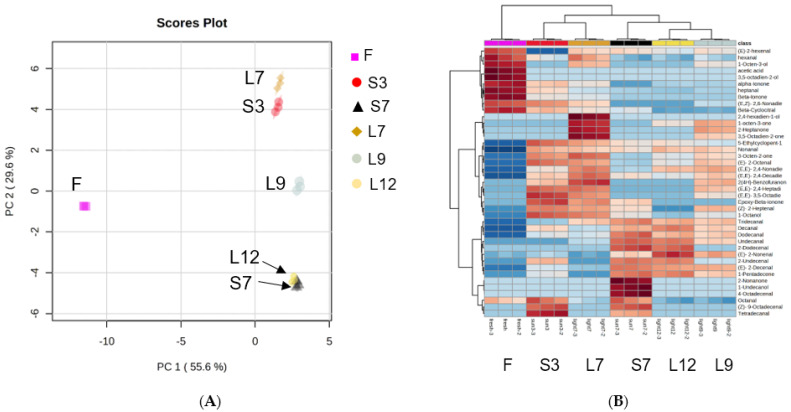
(**A**) Scores plot of volatile compounds *in Gelidium* seaweed.; (**B**) Heatmap of volatile compounds in *Gelidium* seaweed. F (control): Oven-dried *Gelidium* seaweed; S3: Washing and sun drying cycles for 3 times; S7: Washing and sun drying cycles for 7 times; L7: Washing and halogen lamp drying cycles for 7 times; L9: Washing and halogen lamp drying cycles for 9 times; L12: Washing and halogen lamp drying cycles for 12 times.

**Table 1 foods-12-04508-t001:** Agar yields, gelling temperature and hardness of *Gelidium* jelly made by dried red seaweed with different drying processes.

	F	S3	S7	L7	L9	L12
Agar yield (%db)	31.1 ± 6.2 ^bc^	35.6 ± 3.7 ^bc^	45.5 ± 8.7 ^ab^	23.0 ± 10.8 ^c^	58.1 ± 8.5 ^a^	28.1 ± 5.3 ^bc^
Gelling temperature (°C)	26.2 ± 2.3 ^c^	30.8 ± 0.2 ^ab^	31.1 ± 0.6 ^a^	27.5 ± 1.7 ^bc^	29.7 ± 0.7 ^ab^	30.7 ± 0.6 ^ab^
Hardness (N/cm^2^)	0.2 ± 0.01 ^e^	2.7 ± 0.1 ^b^	3.5 ± 0.1 ^a^	2.2 ± 0.1 ^d^	2.4 ± 0.0 ^cd^	2.6 ± 0.2 ^bc^
Springiness	0.1 ± 0.1 ^b^	0.6 ± 0.1 ^a^	0.6 ± 0.2 ^a^	0.4 ± 0.1 ^ab^	0.7 ± 0.3 ^a^	0.6 ± 0.0 ^a^

Different letters (a, b, c, d, e) indicate significant differences at a level of *p* < 0.05. F (control): Oven-dried *Gelidium* seaweed; S3: Washing and sun drying cycles for 3 times; S7: Washing and sun drying cycles for 7 times; L7: Washing and halogen lamp drying cycles for 7 times; L9: Washing and halogen lamp drying cycles for 9 times; L12: Washing and halogen lamp drying cycles for 12 times.

**Table 2 foods-12-04508-t002:** Color parameters of *Gelidium* seaweeds made with different drying processes.

	L*	a*	b*	ΔE ^#^
F	26.1 ± 1.6 ^e^	0.2 ± 0.8 ^c^	15.6 ± 4.3 ^d^	-
S3	50.8 ± 2.6 ^d^	2.4 ± 0.4 ^a^	38.7 ± 1.3 ^c^	28.6 ± 1.8 ^d^
S7	67.9 ± 1.0 ^a^	−2.3 ± 0.1 ^f^	48.5 ± 1.0 ^a^	46.3 ± 2.9 ^a^
L7	49.7 ± 2.6 ^d^	−1.4 ± 0.5 ^e^	36.6 ± 1.2 ^c^	27.0 ± 4.0 ^d^
L9	58.7 ± 2.9 ^c^	1.3 ± 0.5 ^b^	36.9 ± 1.1 ^c^	35.2 ± 2.2 ^c^
L12	63.0 ± 1.9 ^b^	−0.4 ± 0.2 ^d^	44.0 ± 2.8 ^b^	40.7 ± 1.6 ^b^

Expressed as the mean ± standard deviation (n = 3). Values followed by the different letter within each column are significantly different (*p* < 0.05). ^#^ $\Delta E=\sqrt{{\left({\Delta L}^{*}\right)}^{2}+{\left({\Delta a}^{*}\right)}^{2}+{\left({\Delta b}^{*}\right)}^{2}}$. F (control): Oven-dried *Gelidium* seaweed; S3: Washing and sun drying cycles for 3 times; S7: Washing and sun drying cycles for 7 times; L7: Washing and halogen lamp drying cycles for 7 times; L9: Washing and halogen lamp drying cycles for 9 times; L12: Washing and halogen lamp drying cycles for 12 times.

**Table 4 foods-12-04508-t004:** Sensory evaluation of *Gelidium* jelly made by dried seaweeds by different drying processes.

	Appearance	Color	Texture	Flavor	Overall	Fishy Odor *
F	3.35 ± 2.10 ^c^	3.73 ± 2.16 ^c^	2.74 ± 1.96 ^c^	2.27 ± 1.70 ^d^	2.53 ± 1.72 ^c^	7.59 ± 2.25 ^a^
S3	5.65 ± 2.06 ^ab^	6.07 ± 1.67 ^ab^	5.80 ± 2.31 ^a^	4.68 ± 2.08 ^ab^	5.58 ± 2.20 ^a^	4.67 ± 2.50 ^c^
S7	5.87 ± 2.19 ^a^	5.93 ± 2.23 ^a^	6.59 ± 2.00 ^a^	5.38 ± 2.22 ^a^	6.05 ± 1.95 ^a^	3.80 ± 2.39 ^c^
L7	4.67 ± 2.01 ^b^	5.33 ± 1.73 ^b^	2.91 ± 1.48 ^c^	3.02 ± 1.90 ^cd^	3.29 ± 1.76 ^bc^	6.02 ± 2.30 ^b^
L9	5.18 ± 2.00 ^ab^	5.68 ± 1.71 ^ab^	4.09 ± 1.86 ^b^	3.71 ± 1.69 ^bc^	4.00 ± 1.90 ^b^	5.91 ± 2.33 ^b^
L12	5.80 ± 2.10 ^a^	5.90 ± 1.90 ^a^	5.64 ± 2.24 ^a^	4.97 ± 2.20 ^a^	5.44 ± 2.21 ^a^	3.77 ± 2.29 ^c^

Expressed as the mean ± standard deviation (n = 66). Values followed by the different letter within each column are significantly different (*p* < 0.05). Using 9-point hedonic test for appearance, color, texture, flavor, overall preference: 1 = dislike extremely, 5 = neither like nor dislike, and 9 = like extremely. * Using 9-point hedonic test for Fishy odor: 1 = no fishy smell, and 9 = strong fishy smell. F (control): Oven-dried *Gelidium* seaweed; S3: Washing and sun drying cycles for 3 times; S7: Washing and sun drying cycles for 7 times; L7: Washing and halogen lamp drying cycles for 7 times; L9: Washing and halogen lamp drying cycles for 9 times; L12: Washing and halogen lamp drying cycles for 12 times.

## Data Availability

Data is contained within the article or Appendix A.

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
