# Peer review of "Effects of Halogen Lamp and Traditional Sun Drying on the Volatile Compounds, Color Parameters, and Gel Texture of Gongliao Gelidium Seaweed"

_foods, 2023, doi:10.3390/foods12244508_

Round 1

Reviewer 1 Report

Comments and Suggestions for Authors

This study investigated the effect of two drying techniques such as halogen lamps and solar drying, operated in cycles, on the changes in the volatile compounds in Gelidium seaweed. The jelly properties of the dehydrated  Gelidium seaweed were compared. Overall, the study seems interesting.

-The study fits within the scope of the journal.

-The novelty of the study should be established while highlighting the research gap.

-The critical discussion and literature support for the obtained trend or data are missing.

-The drying part of the manuscript should be strengthened.

Following are the other comments.

In the title ‘effect or influence’ word is missing, I believe.

L12: Please delete.

L36-39: Please provide reference.

L41-52: It will be better to have a process flow chart along with suitable references.

L53-54: Please discuss similar research carried out either in the combination of solar or halogen lamp drying and seaweed. Once you derive the research gap, set the objectives.

L63: What was the temperature of seaweed after thawing and before drying?

L67-68: What is the rationale and hypothesis for this step?

L94-95: Present it as an equation.

Sec 2.6: Please mention the reference and write the modifications in that method only.

L139: How did you select these attributes? What do you mean by overall acceptability here?

L154-157: Please delete it accordingly.

L188: What is the possible reason for this trend?

L195: Please revise the statement.

L199-201: What was the drying rate or flux of moisture removal?

L218: Literature support and discussion is missing.

L259: Pls correct the word ‘slope’.

L261: Literature support and discussion is missing.

L271: Is it L value or L* value? Please make it uniform.

L287: Literature support and discussion is missing.

L288: Please make the table captions self-explanatory.

L330: Literature support and discussion is missing.

L339: If the conclusion needs to be made based on Overall acceptability only, then what is the use of other attributes? It is recommended that Overall acceptability should be estimated by combining all the sensory attributes tested. What is the relative importance of each sensory attribute to judge the jelly?

L364-365: This recommendation needs to be elaborated further. What are the reasons for this recommendation?

Author Response

Responses to Comments and Suggestions for Authors

Foods -2715144

Title: Effects of Halogen Lamp and Traditional Sun Drying on the Volatile Compounds, Color Parameters and Gel Texture of Gongliao GelidiumSeaweed

Dear Reviewer #1

Thank you for your instruction on revising abstract, materials and methods, results and discussion. conclusion. We have rewritten the manuscript by Dr. Sung accordingly, and replied to comments and suggestions for authors are listed below point:

Reviewer #1’s comments and suggestions:

Point 1: This study investigated the effect of two drying techniques such as halogen lamps and solar drying, operated in cycles, on the changes in the volatile compounds in Gelidium seaweed. The jelly properties of the dehydrated Gelidium seaweed were compared. Overall, the study seems interesting.

-The study fits within the scope of the journal.

-The novelty of the study should be established while highlighting the research gap.

-The critical discussion and literature support for the obtained trend or data are missing.

-The drying part of the manuscript should be strengthened.

Following are the other comments.

In the title ‘effect or influence’ word is missing, I believe.

Response 1: We have revised the title and texts and added critical discussion and cited literatures in the revised manuscript. Please see the red marked text of the texts of the revised manuscript. Thanks for the suggestion.

Point 2: L12: Please delete.

Response 2: The first sentence of Line 12 was deleted. Sorry for the mistake.

Point 3: L36-39: Please provide reference.

Response 3: Reference #2 was added and thanks for the suggestion. (Please see the revised manuscript in line 38 at page 1)

Point 4: L41-52: It will be better to have a process flow chart along with suitable references.

Response 4: Three references were added to the paragraph of revised manuscript. (Please see the revised manuscript in lines 43 to 55 at pages 1 and 2)

Point 5: L53-54: Please discuss similar research carried out either in the combination of solar or halogen lamp drying and seaweed. Once you derive the research gap, set the objectives.

Response 5: Thanks for this important point. There are plenty of research papers discussing for example the effects of drying methods on volatile compounds of tea, but not seaweeds. The use of halogen lamp for seaweed drying was in my acknowledge the first time in the literature. Phrases of “Sun light in the drying process of Gelidium seaweed was important because it provided bleaching effect on its color. An alternative drying method by halogen lamp instead of sun drying was studied at first time in the literature.” Was added in the revised manuscript at line 58-61. Thanks again for this suggestion.

Point 6: L63: What was the temperature of seaweed after thawing and before drying?

Response 6: Thanks for this informational question. Five hundred grams frozen seaweed for each was sunk in running tap water until completely thawed to tap water temperature (25°C). The sentence was revised in line 72 at page 2 in the revised manuscript.

Point 7: L67-68: What is the rationale and hypothesis for this step?

Response 7: Thanks for the point. Phrases “ In tradition the red seaweed was washed and dried under sun light for 7 cycles. In this study, red seaweed was washed with water, stored at night, dried under sun for 3 and 7 cycles (sun drying), dried by halogen lamp for 7, 9, 12 cycles (halogen lamp drying), respectively. The additional drying cycles of halogen lamp drying (up to 12 cycles) made gel harder.” was added in the revised manuscript at line 77-81.

Point 8: L94-95: Present it as an equation.

Response 8: Thanks for the suggestion. The sentence of color difference was listed as 4 equations in lines 112 to 115 at page 3 of revised manuscript.

Point 9: Sec 2.6: Please mention the reference and write the modifications in that method only.

Response 9: Thanks for pointing out the problems. We added 2 references in this section (2.7). Please see the section 2.7 of revised manuscript.

Point 10: L139: How did you select these attributes? What do you mean by overall acceptability here?

Response 10:  Attributes on appearance, color, flavor, texture, overall acceptability, and fishy odor were selected as the important qualities of Gelidium jellies and asked. Overall acceptability means liking. Thanks for the question. The sentence was revised in line 158-159 of page 4.

Point 11: L154-157: Please delete it accordingly.

Response 11: We are sorry for the mistake again. These two explained sentences were deleted.

Point 12: L188: What is the possible reason for this trend?

Response 12: Thank you for this question. The explanation “ The higher gelling temperature reported by Li et al. [10] might be due to the concen-tration of agar, which applied 2 g of the dried Gracilaria seaweed into less deionized water (100 ml) and more extracting time (2 h). The concentration of agar in this study was ap-plied 1 g of Gelidium seaweed in 100 ml deionized water, and boiled for just 1 h. Therefore, the gelling temperature of G. asiatica was higher in the range of 40°C to 52°C.” was added to lines 211 to 216 at page 5 of revised manuscript. Thanks for mentioning the problem.

Point 13: L195: Please revise the statement.

Response 13: The sentence was revised to “However, the color of red seaweed was not changed.” (Please see line 224 at page 5 of the revised manuscript).

Point 14: L199-201: What was the drying rate or flux of moisture removal?

Response 14: The sentence “All seaweeds dried under sun or halogen lamp for 2 h reaching a moisture content around 10%-13%.” Was added to lines 231 to 232. We are sorry for not evaluating the drying rate of the moisture content of seaweeds during these 2 h drying. We will do that at next year to compare with infrared drying rate and sun drying rate.

Point 15: L218: Literature support and discussion is missing.

Response 15: The sentence was revised to “Table 1 showed Gelidium jelly made from the oven dried sample (control, F) had the weakest gel hardness, which observed a liquid-like characteristic.” In line 253-254. And more discussion was added to previous paragraph. Please see the lines 242 to 247 as red marked texts of the revised manuscript.

Point 16: L259: Pls correct the word ‘slope’.

Response 16: The miss spelling “slop” was revised to “slope”. Thanks for pointing the problem.

Point 17: L261: Literature support and discussion is missing.

Response 17: The literatures #17 and 19 were cited and the discussion of “This indicated the jelly structure of F and L9 groups were deformed during dramatic frequency sweep change. These two Gelidium jellies might remain their jelly structure under mild frequency sweep change. Therefore, F and L9 Gelidium jellies showed a liquid-like jelly.” was added in lines 301 to 304 at page 7 of revised manuscript. Thanks for the suggestions.

Point 18: L271: Is it L value or L* value? Please make it uniform.

Response 18: The color parameters (L, a, b) of Table 2 at page 8 were revised to L*, a*, and b*. Thanks for the corrections.

Point 19: L287: Literature support and discussion is missing.

Response 19: The reference #5 was cited and discussion was added in lines 334 and 335 at page 8.

Point 20: L288: Please make the table captions self-explanatory.

Response 20: The table captions of Table 2 was revised to “Color parameters of Gelidium seaweeds made with different drying processes.” Please see the Table 2 of revised manuscript at page 8.

Point 21: L330: Literature support and discussion is missing.

Response 21: Thanks for this reminder. The sentence “Kim et al. [24] demonstrated β-ionone could be partially removed by UV-photolysis.” Was added in line 383 at page 10.

Point 22: L339: If the conclusion needs to be made based on Overall acceptability only, then what is the use of other attributes? It is recommended that Overall acceptability should be estimated by combining all the sensory attributes tested. What is the relative importance of each sensory attribute to judge the jelly?

Response 22: Several attributes including appearance, color, texture, flavor, and fishy odor and discussion were added at section 3.5 of revised manuscript at page 10-11. Thanks for the great suggestion.

Point 23: L364-365: This recommendation needs to be elaborated further. What are the reasons for this recommendation?

Response 23: Thanks for the suggestions. Sentences “Halogen lamp drying can be applied no matter rain or shine, and it was cleaner and hygienic. The quality of gel produced by halogen lamp-dried seaweed was similar to traditional sun-dried seaweed in the flavor, texture, and color. The power (watts) of the halogen lamp may be higher to speed up the required drying time.” were added at the end of this paragraph in lines 429 and 433 at page 11. We appreciate all the great comments.

The manuscript has been resubmitted to your journal. We look forward to your positive response.

Sincerely yours,

Mingchih Fang, Ph.D.

Associate Professor

Department of Food Science

National Taiwan Ocean University

Reviewer 2 Report

Comments and Suggestions for Authors

Please check the comment file attached

Comments on the Quality of English Language

Overall good academic English just minor typos

Author Response

Responses to Comments and Suggestions for Authors

Foods -2715144

Title: Effects of Halogen Lamp and Traditional Sun Drying on the Volatile Compounds, Color Parameters and Gel Texture of Gongliao GelidiumSeaweed

Dear Reviewer #2

Thank you for your instruction on revising abstract, materials and methods, results and discussion. conclusion. We have rewritten the manuscript by Dr. Sung accordingly, and replied to comments and suggestions for authors are listed below point:

This study entitled ‘Halogen Lamp and Traditional Sun Drying on the Volatile Compounds, Color Parameters and Gel Texture of Gongliao Gelidium Seaweed’ is a great study providing the alternative method of extracting agar with a good yield. The developed technique of using halogen lamp would assist researchers and industry for production of agar-based products. The manuscript is well-written and organized, however, a few issues need to be addressed as mentioned below in comments. Thus, this article should be accepted after a Minor revision. Specific comments are presented below which would help to improve the readability and soundness of the article. Good Luck!

Reviewer #2’s comments and suggestions:

Point 1: Line 12: Please remove this sentence, it is guide only

Response 1: The first sentence of Line 12 was deleted. Sorry for the mistake.

Point 2: Line 23: Please check the grammatical issue in ‘Halogen lamp was found could be an…’

Response 2: The sentence was revised to “Halogen lamp drying could be an alternative processing method for seaweed drying, especially at rainy days.” Please see the last sentence of abstract of revised manuscript at page 1 line 23

Point 3: Line 58: There should be a section for materials about the raw material, chemicals etc.

Response 3: The Materials was added as the section 2.1. (line 64-69) Thanks for the great suggestion.

Point 4: Line 88: Please add how the powder was prepared for color analysis

Response 4: The sentence “Dried red seaweed samples were ground into powder with a pulverizer (D3V-10, Yu-Chi Machinery Co., Ltd., Chan Hua, Taiwan).” Was added to the first sentence of section 2.5. Thanks for the suggestion.

Point 5: Line 89: Sentence need to be rephrased

Response 5: Thanks for the suggestion. The phrase was revised in line 103-106 as “The CIELAB color expressed by L* (lightness), a* (redness/greenness), b* (yellow-ness/blueness) color scale of seaweed powder was conducted with a spectrocolorimeter (TC-1800 MK-II, Tokyo Tokyo, Japan). The illuminant wavelength was 380 to 780 nm. The observer was photomultiplier.” in the revised manuscript.

Point 6: Line 93: Please rephrase the sentence as three time and triplicate are same

Response 6: Thanks for this correction. The phrase was revised to “The seaweed powder was loaded onto a quartz sample cup for each sample and triplicate were recorded per treatment.” In line 109-110 in the revised manuscript.

Point 7: Line 95: Please make the formulae in separate lines

Response 7: Thanks for the suggestion. The sentence of color difference was listed as 4 equations in lines 112 to 115 at page 3 of revised manuscript.

Point 8: Line 101: Lower plate (PP252…parenthesis is missing please fix this

Response 8: The missing parenthesis was added behind the 90°C of the sentence. Thanks for pointing out the problem.

Point 9: Line 153-157: Please remove this statement

Response 9: We are sorry for the mistake again. These two explained sentences were deleted.

Point 10: Line 161-162: What could be the possible reason in agar yield differences between the sun dried and halogen lamp dried samples. Please add some justification.

Response 10: Sentences were added to explain the possible reason in agar yield differences between the sun dried and halogen lamp dried samples. The added sentences were as” It might be due to the repeat washing and sun drying processes would destruct the cell wall and enhance agar extracting yield. Nevertheless, the energy of halogen lamp drying was less than that of sun drying. Therefore, the maximum yield of agar was observed at washing and halogen lamp drying cycles for 9 times (Table 1). However, further washing and halogen lamp drying might leach some available soluble polysaccharides into washing water which made the agar yield decreasing in treatment of 12 times.” Please see the section 3.1 lines 181-187 at page 4 of revised manuscript.

Point 11: Line 186-187: Please suggest the possible reason of lower gelling temperature of agar than US Pharmacopoeia standards?

Response 11: Sentences were added to explain the possible reason of lower gelling temperature of agar than US Pharmacopoeia standards. Please see the section 3.1 lines 211-216 at page 5 of revised manuscript. Thanks for the great suggestion again.

Point 12: Line 191-192: Please suggest the cause of lower hardness of F agar from others?

Response 12: The explanation “It might be due to the water-soluble agar was still hold inside the cell wall in oven drying method without washing and sun drying cycles.” was added in section 3.1 lines 220 to 222 red marked texts at page of revised manuscript.

Point 13: Line 195: Please correct this ‘still was still not changed’

Response 13: The sentence was revised to “However, the color of red seaweed was not changed.” Thanks for pointing out the problem.

Point 14: Line 196: Please check grammar ‘these was not’ should be ‘these were’

Response 14: The grammar mistake was fixed. We appreciate your correction again.

Point 15: Line 259: Please fix the typo in spellings for ‘slop’

Response 15: The wrong spelling word ‘slop’ was revised to ‘slope’. Sorry for making the mistake and thanks for your patient to us.

Point 16: Line 264: Figure 2: labelling ‘B’ and ‘C’ is missing. Please correct accordingly

Response 16: The missing labels (B) and (C) were added to Figure 2 of revised manuscript at page 7. Thanks for pointing out the mistake.

Point 17: Line 287: ‘Extend radiation’ or extended radiation? Please check

Response 17: The phrase ‘extend radiation’ was revised to ‘extended radiation’ as red marked texts in line 333 in the revised manuscript. Thanks again for pointing out the mistake.

Point 18: Line 367: Figure 3. B is missing in the legend, please fix it accordingly

Response 18: We added notes on the caption as well B) in the legend. Thanks again.

The manuscript has been resubmitted to your journal. We look forward to your positive response.

Sincerely yours,

Mingchih Fang, Ph.D.

Associate Professor

Department of Food Science

National Taiwan Ocean University

Reviewer 3 Report

Comments and Suggestions for Authors

The work deals with the drying of seaweed for the production of gels. It compares the traditional solar drying with a halogen lamp.

The work presents a series of errors both in concept and form, and the opportunities for improvement are detailed below:

In l11 avoid the use of gmail type mails, prefer corporate mails.

In general terms the drafting is very deficient, it is necessary that the document is revised and corrected by a native expert in the subject.

There are many errors that come from carelessness in the preparation of the manuscript, for example in several parts of the document parts of the text have been left over the instructions to the authors.

The summary should be completely rewritten, it is not fluent and does not give an idea of the most important aspects and findings of the work.

Clarify what is meant by the term congealed in I37.

It is necessary to write better about the processing of the algae, as it is very convoluted.

In l41 what do you mean by best quality in tradition.

In l43 the superindex is missing.

What is the point of mentioning that red seaweed is produced in Taipei, and also not providing information about specific geographic location and hours of sunshine.

The introduction could be improved by adding information on the advantages and disadvantages of solar drying compared to halogen drying.

In the experimental part it is not clear if the algae were classified to standardize the experiments?

The drying experiments are not clear, it is necessary to revise the wording, on the other hand the conditions of solar drying are very different from halogen drying, also the processing times and stages are not the same so the comparison is not valid. 

It is also not clear how the lux values were measured in the two cases, nor is it clear which was the experimental setup for the halogen lamp drying.

The freeze-drying conditions such as temperature and pressure used are not specified.

In l80 it is not specified to which temperature corresponds hot?

It is necessary to specify how many were the replicates for each one of the measurements, in some parts if it is shown but not in all.

In the color measurements it is necessary to inform which was the illuminant and the observer used for the measurements.

Use the equation editor for the color formulas.

L154-157 delete text from the instruction for authors.

In general terms the discussion was shallow and unorganized.

It is not clear what UV-C was used for in l194 and this should go in the materials and methods section.

It is also not clear what the characteristics of the halogen were, was it an infrared lamp, power, dimensions, etc.

The experimental design is not clear, you could put a table with the experiments and/or maybe some explanatory scheme of the experimental process that the algae received.

In table 2 change L a b by L* a* b*.

It would be interesting to relate the results of the sensory analysis with the volatiles measurements.

The conclusion should be improved with emphasis on the generation of knowledge and its possible scientific and also practical applications.

Author Response

Responses to Comments and Suggestions for Authors

Foods -2715144

Title: Effects of Halogen Lamp and Traditional Sun Drying on the Volatile Compounds, Color Parameters and Gel Texture of Gongliao GelidiumSeaweed

Dear Reviewer #3

Thank you for your instruction on revising abstract, materials and methods, results and discussion. conclusion. We have rewritten the manuscript by Dr. Sung accordingly, and replied to comments and suggestions for authors are listed below point:

Reviewer #3’s comments and suggestions:

The work deals with the drying of seaweed for the production of gels. It compares the traditional solar drying with a halogen lamp.

 The work presents a series of errors both in concept and form, and the opportunities for improvement are detailed below:

Point 1: In l11 avoid the use of gmail type mails, prefer corporate mails.

Response 1: We have replaced the school email address of Mr. Liao, 11032034@mail.ntou.edu.tw to his gmail address. Please see the revised institutional email at page 1 line 11.

Point 2: In general terms the drafting is very deficient, it is necessary that the document is revised and corrected by a native expert in the subject.

 There are many errors that come from carelessness in the preparation of the manuscript, for example in several parts of the document parts of the text have been left over the instructions to the authors.

 The summary should be completely rewritten, it is not fluent and does not give an idea of the most important aspects and findings of the work.

Response 2: We have rewritten and rechecked the texts and abstract in our article carefully as red marked texts in the revised manuscript. Thanks for the suggestions and we very much appreciate your consideration on this matter. (Please see the revised manuscript).

Point 3: Clarify what is meant by the term congealed in I37.
Response 3: The word “congealed” was revised to “gelled”. Thanks for the great comment. Please see the revised introduction at page 1 line 37.

Point 4: It is necessary to write better about the processing of the algae, as it is very convoluted.

Response 4: The processing of the algae is complicated and it takes a lot of labor. We added the cited reference to the second paragraph of introduction of the revised manuscript at page 2 lines 51-55. Thanks for the comment.

Point 5: In l41 what do you mean by best quality in tradition.

Response 5: The phrase “best quality” was revised to “desired gel strength as the soft agar pudding”. Please see the introduction section of the revised manuscript at page 1 lines 41-42. Thanks for the great comment.

Point 6: In l43 the superindex is missing.

Response 6: The superindex is fixed. Please see the introduction of revised manuscript line 43 at page 1. Thanks for pointing out the problem.

Points 7: What is the point of mentioning that red seaweed is produced in Taipei, and also not providing information about specific geographic location and hours of sunshine.

Response 7: The Gongliao district of New Taipei, Taiwan is the main production area of Gelidium seaweed since several decades ago. And the seaweeds were exported to Japan to make agar. We added the geographical coordinate of Gongliao. The illuminance of this area is high enough during summer time except typhoon days. Please see the introduction of the revised manuscript in lines 42 to 46 at pages 1 and 2.

Point 8: The introduction could be improved by adding information on the advantages and disadvantages of solar drying compared to halogen drying.

Response 8: The explanatory texts of the advantages and disadvantages of solar drying compared to other food drying methods were added at the introduction of revised manuscript page 1 lines 51-53 to specify the development of the new drying process. Thanks for the great suggestion.

Point 9: In the experimental part it is not clear if the algae were classified to standardize the experiments?

Response 9: Thank for the suggestion. Three batches of algae were processed and the phrase was added at the Materials and Method sections at page 2 lines 64 and 69 of the revised manuscript.

Point 10: The drying experiments are not clear, it is necessary to revise the wording, on the other hand the conditions of solar drying are very different from halogen drying, also the processing times and stages are not the same so the comparison is not valid. 

Response 10: We have added several sentences to explain the drying experiments for sun drying, halogen lamp drying, oven drying and UV light drying. Although the drying temperature might not be different. The final moisture content of algae was around 10-13% for storage. Please see the section 2.2 of the revised manuscript at page 2.

Point 11: It is also not clear how the lux values were measured in the two cases, nor is it clear which was the experimental setup for the halogen lamp drying.
Response 11: We put the illuminometer on the surface under sun drying algae and LED halogen drying lamp to evaluate the lux value and thermometer in the center of algae to evaluate the temperature of sun drying and halogen lamp drying. We have added several phrases to explain the experimental setup. Please see the revised section 2.2 at page 2 lines 73-76. If it is not clear enough, we would revised the texts again Thanks for the comments.

Point 12: The freeze-drying conditions such as temperature and pressure used are not specified.

Response 12: The freeze-drying conditions for agar content calculation were added. Thanks for the comments. (Please see the Materials and Methods section 2.3 of the revised manuscript at page 2 lines 89-90).

Point 13: In l80 it is not specified to which temperature corresponds hot?
Response 13: The sentence was revised to “The above extracted and filtered agar solution each was immediately poured into a glass beaker (diameter of 38.0 mm and height of 38.0 mm).” Thanks for the great comment. Please see the revised sentence at page 2 lines 94-95.

Point 14:  It is necessary to specify how many were the replicates for each one of the measurements, in some parts if it is shown but not in all.

Response 14: We added “All tests were performed in triplicate. “ at section 2.9 of Materials and Methods of the revised manuscript at page 2 line 92 and page 3 line 110. Thanks for the suggestion.

Point 15:  In the color measurements it is necessary to inform which was the illuminant and the observer used for the measurements.

Response 15: The illuminant wavelength was 380 to 780 nm. The observer was photomultiplier. These two sentences were added at section 2.5 lines 107 to 108. And The color parameters (L, a, b) of Table 2 at page 8 were revised to L*, a*, and b*. Thanks for the great comment.

Point 16: Use the equation editor for the color formulas.

Response 16: Thanks for the suggestion. The sentence of color difference was listed as 4 equations in lines 112 to 115at page 3 of revised manuscript.

Points 17:  L154-157 delete text from the instruction for authors.

Response 17: We are sorry for the mistake again. These two explained sentences were deleted.

Point 18:  In general terms the discussion was shallow and unorganized.

Response 18: We added more discussion and reorganized them in the texts. Please see the Results and Discussion section 3 of the revised manuscript at pages 4 to 12.

Point 19: It is not clear what UV-C was used for in l194 and this should go in the materials and methods section.

Response 19: We have added several sentences to explain the drying experiments for sun drying, halogen lamp drying, oven drying and UV light drying. Although the drying temperature might not be different. We try to evaluate whether the UV-C light could bleach the red seaweeds. However, it is not work! The final moisture content of algae was around 10-13% for storage. Please see the section 2.2 of the revised manuscript at page 2.

Point 20: It is also not clear what the characteristics of the halogen were, was it an infrared lamp, power, dimensions, etc.

Response 20: We added the dimension of halogen lamp in line 73 at page 2. Thanks for the comment.

Point 21: The experimental design is not clear, you could put a table with the experiments and/or maybe some explanatory scheme of the experimental process that the algae received. Response 21: Three batches of algae were processed and the phrase was added at the Materials and Method sections at page 2 lines 64 to 69 of the revised manuscript. We have added several sentences to explain the drying experiments for sun drying, halogen lamp drying, oven drying and UV light drying. Although the drying temperature might not be different. The final moisture content of algae was around 10-13% for storage. Please see the section 2.2 of the revised manuscript at page 2.

Point 22: In table 2 change L a b by L* a* b*.

Response 22: The color parameters (L, a, b) of Table 2 at page 8 were revised to L*, a*, and b*. Thanks for the great comment. (Please see the Table 2 of the revised manuscript).

Point 23: It would be interesting to relate the results of the sensory analysis with the volatiles measurements.
Response 23: It is great idea to correlate the sensory result with identified volatile compounds and concentrations. However, this study only conducted appearance, color, texture, flavor, fishy odor and overall acceptability as attributes. There are only flavor attribute and fishy odor related to volatile. The sensory evaluation is too simple and we think there is no need to conduct the correlation between flavor attribute and volatiles. Thanks for the suggestion.

Point 24: The conclusion should be improved with emphasis on the generation of knowledge and its possible scientific and also practical applications.

Response 25: The conclusion was revised by adding the generation of knowledge and its possible scientific and practical applications. Thanks for the great comment.

The manuscript has been resubmitted to your journal. We look forward to your positive response.

Sincerely yours,

Mingchih Fang, Ph.D.

Associate Professor

Department of Food Science

National Taiwan Ocean University

Round 2

Reviewer 3 Report

Comments and Suggestions for Authors

The authors considered most of the suggestions and therefore the article improved considerably.

The error about colour measurement still persists, it is necessary to specify correctly which illuminant and observer were used (e.g. D65 10°, check the manual of your equipment and/or topics about colour theory).

The format of some references needs to be revised.

Author Response

Dear reviewer

Thank you for your time and patience. The error about color measurement was corrected as "CIE D65 illuminant and 10o Standard Observer were used." Thank you for the correction. Actually, I did not know about the illuminant and observer before. Thanks for your kindly correction, and I learned the color theory. I also checked the format of references. Please see the revised manuscript. Thanks a lot for your review and suggestions.

Sincerely, Mingchih Fang